# Effectiveness of Therapeutic Patient Education Interventions in Obesity and Diabetes: A Systematic Review and Meta-Analysis of Randomized Controlled Trials

**DOI:** 10.3390/nu14183807

**Published:** 2022-09-15

**Authors:** Jorge C. Correia, Ahmed Waqas, Teoh Soo Huat, Karim Gariani, François R. Jornayvaz, Alain Golay, Zoltan Pataky

**Affiliations:** 1Unit of Therapeutic Patient Education, WHO Collaborating Centre, Division of Endocrinology, Diabetology, Nutrition and Therapeutic Patient Education, Geneva University Hospitals and University of Geneva, 1206 Geneva, Switzerland; 2Institute of Population Health, University of Liverpool, Liverpool L69 7ZA, UK; 3Department of Community Health, Advanced Medical & Dental Institute, Universiti Sains Malaysia, 13200 Kepala Batas, Penang, Malaysia

**Keywords:** diabetes mellitus, obesity, meta-analysis, patient education, self-management, disease management

## Abstract

Diabetes mellitus (DM) and obesity account for the highest burden of non-communicable diseases. There is increasing evidence showing therapeutic patient education (TPE) as a clinically and cost-effective solution to improve biomedical and psychosocial outcomes among people with DM and obesity. The present systematic review and meta-analysis present a critical synthesis of the development of TPE interventions for DM and obesity and the efficacy of these interventions across a range of biomedical, psychosocial and psychological outcomes. A total of 54 of these RCTs were identified among patients with obesity and diabetes and were thus qualitatively synthesized. Out of these, 47 were included in the quantitative synthesis. There was substantial heterogeneity in the reporting of these outcomes (I^2^ = 88.35%, Q = 317.64), with a significant improvement noted in serum HbA1c levels (standardized mean difference (SMD) = 0.272, 95% CI: 0.118 to 0.525, *n* = 7360) and body weight (SMD = 0.526, 95% CI: 0.205 to 0.846, *n* = 1082) in the intervention group. The effect sizes were comparable across interventions delivered by different modes and delivery agents. These interventions can be delivered by allied health staff, doctors or electronically as self-help programs, with similar effectiveness (*p* < 0.001). These interventions should be implemented in healthcare and community settings to improve the health outcomes in patients suffering from obesity and DM.

## 1. Introduction

Obesity and diabetes mellitus (DM) account for the highest burden of non-communicable diseases. Recent meta-analyses revealed that the prevalence of central obesity globally is around 41.5% (95% CI 39.9–43.2%) using pooled data from 288 studies with over 13 million participants [1]. Obesity is also highly comorbid with type 2 DM with a prevalence of 8.5% among adults around the globe [2]. The public health and socioeconomic impacts of these disorders are immense. Reports by the American Diabetes Association estimate the economic costs of the DM at USD 327 billion including the direct medical costs and indirect productivity costs [3]. The costs of obesity in the US alone are estimated at USD 1.72 trillion including USD 480.7 billion in direct healthcare costs and USD 1.24 trillion in indirect costs [4]. The high prevalence, morbidity and mortality and socioeconomic costs associated with these metabolic disorders warrant innovative solutions to deliver sustainable and equitable healthcare across the globe.

Studies have shown the impact of therapeutic patient education (TPE) as a clinically and cost-effective solution to improve biomedical and psychosocial outcomes among people with metabolic disorders [5,6,7,8,9]. The primary aim of TPE is to help people with different disorders understand the nature of their disease and empower them with knowledge and skills. Thus, TPE can help them make informed decisions, self-manage their symptoms and prevent further complications [10]. TPE is also important to develop an effective therapeutic alliance between the patients and caregivers and enables a more collaborative approach to treatment [10]. This is particularly important because the inclusion of the patients as decision-makers and stakeholders in their treatment improves attitudes and practices, self-efficacy and adherence, which are important mediators of improved biomedical and psychosocial outcomes [11].

The efficacy of TPE interventions has been established in several randomized controlled trials [6,12,13]. Although recently published systematic reviews and meta-analysis have estimated their effectiveness, these have either been limited to a specific set of TPE interventions or subsets of patients [12,14,15]. Moreover, none of these has critically reviewed the use of TPE interventions in psychosocial and psychological outcomes or the theoretical underpinning and implementation considerations for these interventions in different settings. Our systematic review and meta-analysis address this gap in knowledge and aim to:
(a)Present a critical synthesis of the theoretical basis and development of TPE interventions for obesity and diabetes.(b)Present quantitative evidence for the efficacy of these interventions across a range of biomedical, psychosocial and psychological outcomes.

This review is part of a larger project PARTNERSHIP (**P**utting the p**A**tient fi**R**s**T**: ma**N**agem**E**nt of ch**R**onic di**S**eases by t**H**erapeut**I**c **P**atient education), leading a series of evidence synthesis studies on the role of therapeutic patient education in the management of chronic disorders.

## 2. Materials and Methods

### 2.1. Information Sources and Search Strategy

The present systematic review and meta-analysis (SRMA) build on our previous large-scale SRMA, which presented the effectiveness of TPE interventions across all medical specialties [16]. The protocol for the large-scale SRMA was registered a priori in PROSPERO (CRD42019141294). By using a subset of the studies from the parent SRMA, this SRMA provides a more in-depth review and critical analysis of TPE interventions for obesity and diabetes. This SRMA follows the reporting guidelines recommended by the PRISMA guidelines for reporting systematic reviews [17]. The database search for the systematic review was performed in Web of Science, MEDLINE, CINAHL, PsycINFO and COCHRANE databases, from inception until August 2019. The search strategy for the original systematic review is presented in Appendix A.

### 2.2. Eligibility Criteria

For this review, we considered all those studies which presented the effectiveness of TPE interventions in chronic metabolic disorders (obesity and diabetes mellitus type-I and type-II) presenting in community or healthcare settings. We considered only randomized and cluster randomized controlled trials conducted among adults ≥ 18 years old. We considered a range of outcomes including but not limited to biological parameters, psychological symptomology and quality of life (QoL) indicators. These indicators may include (but are not limited to): disease progression, treatment outcome, rate of complications, rate of relapse, hospitalization, self-care, compliance and adherence to treatment, health knowledge, attitudes and behaviors, and QoL assessed using valid and reliable scales. Only primary outcomes tested at primary time points were included.

### 2.3. Data Extraction

Two reviewers working independently from one another screened the articles for eligibility and performed the data extraction. In case of discrepancies, senior investigators were involved to arrive at a final decision. Using a pretested data extraction form, we extracted qualitative data on the interventions which included the rationale of interventions, delivery techniques and content of interventions and density of dose and characteristics of delivery agents. We also extracted data on modalities used for the delivery of TPE interventions. An effort was made to map the content and syllabus of each intervention using a framework developed by the review team. The content of the interventions was mapped to five domains: disease management, lifestyle changes, coping skills, disease processes and interpersonal skills.

### 2.4. Meta-Analysis

Quantitative data to calculate effect sizes included mean (SD) and sample sizes of intervention and control arms for continuous outcomes and frequency of events and sample sizes for categorical outcomes. If these data were not available, then we extracted odds ratios, mean differences and sample sizes [18]. Standardized mean differences (SMD) were calculated for continuous outcomes. Data were pooled using random effects due to expected methodological and clinical heterogeneity across the studies [18]. A funnel plot was used to assess publication bias in reporting outcomes, after which Duval and Tweedie’s trim and fill method would be used to provide adjusted effect sizes and associated 95% CI [19]. A subgroup analysis with mixed effects and meta-regression analyses were performed to delineate moderators of TPE intervention effects [20] where moderators were reported in at least four and ten studies, respectively. Subgroup analyses were run to identify differences in the effect sizes of TPE interventions according to the type of delivery agents and mode of delivery. A random effects meta-regression analysis was used to analyze associations between the effect sizes of TPE interventions with the content of interventions (disease management, lifestyle changes, coping skills, disease processes and interpersonal skills).

### 2.5. Risk of Bias Assessment

The Cochrane tool for risk of bias assessment in RCTs was used to assess the risk of bias in the selection and allocation of study participants to interventions, blinding of outcome assessors, attrition bias and other biases [18,21].

## 3. Results

A careful review of 5388 titles and abstracts was performed to identify 984 full texts for eligibility for inclusion in the review. Out of these 984 full texts, we included 497 in our original database of TPE interventions across all medical specialties. A total of 54 of these RCTs were conducted among patients with obesity and DM and were thus qualitatively synthesized. Out of these, 47 were included in the quantitative synthesis (Figure 1). Seven studies did not provide sufficient data for the meta-analysis, and therefore, were synthesized narratively. The reasons for the exclusion of studies are provided in more detail elsewhere [22].

### 3.1. Characteristics of Interventions

Out of the 54 eligible interventions, 46 (85.19%) were focused on DM, and six (11.11%) were on obesity and overweight. These interventions were delivered by allied health workers (*n* = 28), multidisciplinary teams (*n* = 17), research teams (*n* = 6), peers and peer leaders (*n* = 2) and doctors (*n* = 1). A variety of delivery formats were used including in groups (*n* = 16), individually (*n* = 15), electronically (*n* = 8) and mixed formats (*n* = 12). These interventions varied in the use of delivery techniques with the highest proportion of interventions facilitating supervision (*n* = 54), interactive presentations (*n* = 51), practical work (*n* = 44), use of information media (*n* = 33), round table discussions (*n* = 20), brainstorming (*n* = 14), use of logbooks (*n* = 11) and animation media (*n* = 11) among others (Figure 2). Detailed characteristics of the included RCTs are presented in Appendix A.

### 3.2. Ingredients of Interventions

The most taught components for disease management included managing complications (*n* = 48) and self-monitoring (*n* = 41). Information for lifestyle changes spanned across the prevention of complications (*n* = 49), implementation of lifestyle changes (*n* = 45) and awareness of risk factors (*n* = 38). Several cognitive and behavioral coping skills were also taught in these interventions including self-care (*n* = 52), situational awareness (*n* = 45), critical thinking (*n* = 40), goal setting (*n* = 35) and self-confidence (*n* = 31), among others. Information regarding disease processes revolved around health behaviors (*n* = 51), and interpersonal skills were taught in 19 interventions (Figure 3). There were no differences in the use of different curriculum content of the TPE interventions based on varying delivery formats (Appendix A).

### 3.3. Description of Interventions According to Delivery Format

#### 3.3.1. Electronic Interventions

Among these interventions, five were delivered through internet media [23,24,25,26,27] and three using telephones [28,29,30]. Carter et al. tested an online diabetes self-management intervention for urban African Americans with type 2 DM, to enable them to assume more responsibility for their health and improve DM-related outcomes [23]. This intervention was delivered by telehealth nurses who delivered biweekly 30 min video conferences including modules on self-management, nutrition education and physical activity. It also allowed for social networking among patients. Mckay et al. aimed to improve physical activity among DM patients with sedentary lifestyles mediated by occupational therapists. The participants were provided the intervention in a group setting, received and could post messages to an online personal coach and participated in peer group support areas [24]. Shea et al. tested an intervention based on videoconferencing and remote monitoring of glucose and BP facilitated by a project case manager under the supervision of diabetologists [27]. Blomfield et al. tested two interventions: (i) guided self-help strategies with a website-enabled online food and exercise diary with feedback provided; and (ii) in this arm, participants were provided with DVDs, weight loss handbooks, a pedometer and lifestyle diary with no feedback. Ramadas et al. utilized a web-based dietary lesson plan personalized according to patients and supervised by study nutritionists [26].

The telephone-delivered interventions prevented glycemic relapse through routine follow-ups ensuring self-care behaviors [28], adherence to medication using educational modules [30] and automated calls to improve the management of type 2 DM (using a Bluetooth-enabled glucometer) by improving physical activity, medication-taking and nutrition [29].

#### 3.3.2. Group Interventions

Interventions were delivered to groups of patients as small as 3 participants [31] and as large as 70 [32]. Seven interventions were delivered by allied health workers including technical health assistants [33,34], educators [35,36], social workers [37], volunteer peers managed by diabetes nurses [38] and psychologists [39].

Kruger et al. employed technical assistants trained by nurses and podiatrists. Participants were provided with lecture videos and a foot check sheet, along with the usual teaching on foot care and hands-on learning of foot care techniques. Chaveepojnkamjorn et al. developed a self-help program facilitated by trained allied health staff who imparted knowledge of DM, skills for dietary control and physical exercise, self-monitoring and motivation for experience sharing and training skills for group leaders [34]. Trained educators were utilized to facilitate education programs for physicians and people with diabetes where active patient participation and self-care were emphasized to improve DM care [35]. Gillet et al. provided structured education to newly diagnosed patients with DM to aid in goal-setting mainly for weight loss and smoking cessation [36].

Soennichsen et al. delivered modules of patient education on interdisciplinary care where at regular intervals agreement on therapeutic goals and shared patient–physician decision-making were encouraged [31]. Three of these interventions [37,38,39] utilized psychotherapeutic skills: problem-solving, provision and social and emotional support and self-regulation (dietary and weight management) to improve diabetes and obesity outcomes.

By utilizing multidisciplinary teams, five of the interventions focused on imparting self-care and self-management techniques for DM [7,9,40,41,42]. Two interventions used psychotherapeutic skills. Trouilloud et al. imparted skills in dietary management, physical activity and medication using educational and problem-solving activities [7]. Whitehead et al. used principles of mindfulness and acceptance training about difficult thoughts and feelings [9]. In other interventions delivered by multidisciplinary teams [40,41,42], unique strategies included the inclusion of family members [40], and training to recognize when BG is too high or low and anticipate when it is likely to rise or fall [41].

Interventions delivered by trained peers [43] and peer leaders [44] focused on the educational and self-management aspects of diabetes. Interventions delivered by research teams, besides imparting basic education and self-management skills, also touched upon more technical points on the kinetics of insulin and cues signaling hypoglycemia and management of medicine [32,45].

#### 3.3.3. Individually Delivered Interventions

There were 16 interventions which aimed to deliver TPE interventions using a personalized and individual approach [5,46,47,48,49,50,51,52,53,54,55,56,57,58,59,60]. All these interventions were focused on lifestyle and disease education and self-management techniques for diabetes except for four interventions which were primarily aimed at smoking cessation among patients with diabetes [48,50], personalized diabetes risk assessments during ophthalmologic visits [58] and improving physical activity [60].

Six of these interventions employed clinical examinations and assessments of individual patients before delivery of the TPE intervention [5,47,50,52,54,60]. For instance, Moriyama et al. performed a diabetes-related clinical examination for individual patients followed by education about diet, exercise, smoking cessation, medication and stress management and the prevention of diabetic complications [5]. Nejhaddadgar et al. after performing clinical assessments delivered a program based on the PRECEDE model (knowledge, attitudes, social and family support), to improve self-efficacy [47]. Shubayama et al. performed a one-to-one assessment of eating patterns, physical activity and self-care for diabetic complications, and thereafter assisted in goal-setting and regular evaluation and feedback support [52]. Besides offering educational modules, Seligman et al. provided food packages containing diabetes-appropriate foods and HbA1c testing. Aiello et al. offered an intervention package for ophthalmological care for patients with DM including measurement of HbA1c level, blood pressure and retinopathy severity; demonstration of a graph showing the risk for worsening retinopathy and comparing previous and current HbA1c levels; risk assessment for renal disease and retinopathy [58].

#### 3.3.4. Multimodal Interventions

Among these 13 interventions, TPE was provided by pharmacists [13,61,62,63], nutritionists [64], counselors [65], lifestyle coaches [66], multidisciplinary teams [67,68,69,70] and researchers [71,72]. Pharmacists delivered their interventions either using face-to-face meetings or by telephoning the patients. These interventions were focused on medication counselling but also included preventive education. The pharmacist-led interventions addressed identification and resolution of drug-related problems and adherence to medication and demonstration of insulin injection technique [61], generalized education on DM and meal planning using a food pyramid chart provided by the pharmacists [62] and initial face-to-face sessions followed by telephone calls [63]. Sonal Sekhar and colleagues also tested the effectiveness of clinical pharmacists in providing education on foot care and podiatry reviews [13].

Three interventions tested personalized care for patients with obesity [64,65,66]. These interventions were personalized according to the patient’s needs and had varied goals. Assuncao et al. employed two nutritionists to aid patients in reducing weight and controlling risk factors of noncommunicable chronic diseases. The intervention recipients received a manual with photographs of the portion sizes of prescribed foods; dietetic prescriptions; guidance on choosing and substituting foods; encouragement to consume vegetables, fruit and low-fat foods; and encouragement to perform physical activity and promotion of follow-up visits [64]. Perri et al. initiated obesity interventions in underserved rural settings using counsellors to encourage sustained weight loss [65]. Wadden et al. tested a 2 year-long intervention where lifestyle coaches performed quarterly visits combined with monthly 10 to 15 min long sessions followed by telephone-delivered counselling every other month in year 2. Besides this, patients also received a pedometer, calorie-counting book, dietary and physical activity goal-setting, meal replacements or weight-loss medication [66].

TPE interventions delivered by multidisciplinary teams were tested in four interventions [67,68,69,70]. These interventions were quite heterogeneous. Korhonen et al. delivered the TPE interventions on DM to hospitalized patients with instructions to adjust the insulin dose in special situations [67]. Wagner et al. utilized automated telephone self-management and patient activation linked to nursing care by phone [70]. Chao et al. tested an integrated health management model to improve the health of older adults with DM in the community by ensuring health record establishment, health evaluation and health management [69]. The remaining two interventions were delivered by researchers [71,72] who tested TPE interventions comprising structured education for functional insulin treatment [72], and computer-assisted intervention providing automated feedback on key barriers to dietary self-management, goal-setting and problem-solving counselling [71].

### 3.4. Efficacy of Interventions

#### 3.4.1. Biological Outcomes

A total of 31 studies (37 trial arms, *n* = 9879 participants) on the effectiveness of therapeutic patient education interventions were included in meta-analyses.

A total of 25 studies recruiting patients with diabetes, reported HbA1c serum levels as a primary outcome. There was substantial heterogeneity in the reporting of this outcome (I^2^ = 88.67%, Q = 203.06, *p* < 0.001), with significant reductions in serum HbA1c levels noted among the intervention group (SMD = 0.272, 95% CI: 0.118 to 0.525, *n* = 7360) (Figure 4). Publication bias was not evident in the reporting of this outcome (Egger’s regression *p* = 0.59, Figure 5). As per subgroup analyses, effect sizes for these interventions differed significantly with delivery agents. However, no variation in effect sizes was evident based on the format of delivery (Q = 2.28, *p* = 0.52). The intervention delivered by multidisciplinary teams yielded the highest effect sizes (SMD = 0.35, 95% CI: 0.09 to 0.61) followed by allied health workers (SMD = 0.26, 95% CI: 0.10 to 0.43) (Table 1). No associations were found between the content of interventions and their effect sizes (Table 2).

eGFR was reported among patients with diabetes in two studies only. There was no evidence for statistical heterogeneity (I^2^ = 0%, Q = 0.18, *p* = 0.67). A fixed-effects meta-analysis yielded large effect sizes (SMD = 0.795, 95% CI: 0.573 to 1.016, *n* = 340). Serum glucose levels were reported in only one study, showing large effect sizes in favor of the intervention group (SMD = 1.144, 95% CI: 0.848 to 1.439, *n* = 485). UKPDS did not exhibit significant improvements as reported in only one study (SMD= −0.138, 95% CI: −0.391 to 0.115, *n* = 240).

Seven studies reported a change in body weight as an outcome among patients with diabetes. There was significant evidence of heterogeneity (I^2^ = 80.06%, Q = 35.42, *p* < 0.001). Significant improvements (Figure 6) were noted in body weight among recipients of TPE interventions (SMD = 0.526, 95% CI: 0.205 to 0.846, *n* = 1082). No publication bias was evident in the visualization of the funnel plot (Figure 7) and statistically through Egger’s regression (*p* = 0.60). Effect sizes varied according to both delivery agents (Q = 15.02, *p* < 0.001) and the format of delivery (Q = 19.61, *p* < 0.001). The highest effect sizes were noted for interventions delivered by allied health workers (SMD = 0.44, 95% CI: 0.11 to 0.77) and for interventions delivered through the internet (SMD = 1.40, 95% CI: 0.94 to 1.87) and in a mixed format (SMD = 0.51, 95% CI: 0.06 to 0.96).

Improvements in BMI were also noted among patients with obesity in two interventions (SMD = 0.358, 95% CI: 0.134 to 0.581; I^2^ = 0%).

#### 3.4.2. Adherence

Adherence to treatment regimen was reported in two studies, with a cumulative sample size of 521 participants. There was no significant statistical heterogeneity in the reporting of this outcome (I^2^ = 0%, Q = 0.01). It yielded a weak and imprecise effect size in favor of the intervention group (SMD= 0.310, 95% CI: 0.05 to 0.57).

#### 3.4.3. Knowledge

Knowledge was reported as a primary outcome in two studies, with a cumulative sample size of 199 participants. The reporting of this outcome was substantially heterogeneous (I^2^ = 98.66%, Q = 74.36). Although the effect size showed improvement in favor of the intervention group, the effect sizes were imprecise and statistically non-significant (SMD = 2.60, 95% CI: −1.44 to 6.64).

#### 3.4.4. Quality of Life

QoL-mental was reported in two studies, with a cumulative sample size of 255 participants. It did not reveal a statically significant improvement in favor of the intervention group (SMD = 1.57, 95% CI: −0.54 to 3.68, I^2^ = 98.05%, Q = 51.32). QoL-physical was reported in only three studies, with a cumulative sample size of 410 participants. There was evidence for high statistical heterogeneity with no evidence of improvement among participants undergoing TPE interventions (SMD = 0.682, 95% CI: −0.16 to 1.52); however, the effect sizes were imprecise.

#### 3.4.5. Risk of Bias

Among the included RCTs, the risk of bias was low in selective reporting (*n* = 51), attrition bias (*n* = 38), random sequence generation (*n* = 20) and allocation concealment (*n* = 5) (Figure 8).

## 4. Discussion

This systematic review and meta-analysis critically analyze the experimental literature on TPE interventions for obesity and diabetes. Using piloted taxonomies, we present a synthesis of delivery techniques and modalities adopted by various investigators. Besides this, the curriculum and skills covered in each intervention have been summarized to aid in the future development of TPE interventions. We show that TPE interventions bring about significant improvements in biomedical outcomes among patients with DM and obesity. Only a few of the interventions explored psychological and psychosocial outcomes or mediators of TPE interventions as primary goals.

The present systematic review and meta-analysis demonstrate the moderate strength of effect sizes in the improvement of biomedical outcomes among patients with DM and obesity. It corroborates findings from multiple RCTs and meta-analyses reporting TPE interventions as an essential and effective component of patient care [73,74,75]. A plethora of literature on TPE indicate these interventions to be a core tenet in building trust and a therapeutic relationship between the physician and the patient [10,76,77].

We could not find significant differences in the QoL of participants undergoing TPE interventions. However, our analyses are inconclusive as we only considered data from two studies with 255 participants and reported QoL as a primary outcome. The previous literature shows that TPE interventions also enable health professionals to tackle the psychosocial aspects (including QoL, depression and anxiety) of chronic diseases. For instance, people with obesity, in addition to their symptoms, also demonstrate fear, loneliness and stigma which may mediate food intake and determine future prognosis. The TPE approach tackles these biopsychosocial challenges to achieve holistic health. Recognizing its importance, stakeholders in the field of TPE have long advocated for improving the competency of physicians and allied health staff in delivering TPE [10].

The World Health Organization has identified several barriers to the implementation of TPE interventions [10], including a lack of human resources trained in TPE. Another major obstacle is the nature of medical training which results in a pervasive and mechanical approach to the treatment of patients. This approach to medical training is often suitable for acute diseases, but managing chronic diseases requires a more holistic approach. There is insufficient teamwork between the physicians, allied health staff and community stakeholders. This resistance to teamwork is often counterintuitive from an implementation perspective of TPE. There is a lack of commitment from policymakers and institutions who believe in the biomedical approach to medical training. This often translates to a lack of educational resources, finances and infrastructure necessary for the implementation of TPE interventions on a large scale [10].

The present systematic review reveals that TPE interventions delivered through different media and delivery formats may be equally effective. Similarly, trained allied health staff may present a more cost-effective solution to establishing a TPE program in hospital and community settings [9,39,52,70,78]. Therefore, these interventions can be tailored to the setting according to the availability of human and financial resources. Some of these interventions could be more personalized and involve the provision of expensive hardware for self-monitoring and management. This may not be possible in low-resourced settings. Nonetheless, by recognizing the needs of the end-users and the acceptability of TPE interventions, these could be tailored.

Another consideration is the multidisciplinary nature of TPE interventions. These interventions may be underpinned by different theories and psychotherapeutic underpinnings [11]. The psychotherapeutic approaches include principles of cognitive and behavioral therapies, learning theories and different definitions and meanings of health literacy. More influential theories include Bandura’s social foundations of thought and action based on social cognition theory and Ajzen’s theory of planned behavior [11]. However, we show that where educational and human resources do not allow, simpler programs may also be equally effective. This assertion is corroborated by our meta-regression analyses where the use of different curriculum content and techniques does not lead to variation in the effect sizes of TPE interventions.

## 5. Strengths and Limitations

The present systematic review and meta-analysis is the first concerted effort to synthesize evidence for the development and implementation of TPE interventions. Besides these important considerations, the efficacy of these interventions in obesity and diabetes has been thoroughly analyzed. However, there are also some weaknesses in the study design. The studies included in this systematic review were identified from a subset of studies from our larger review on 497 TPE interventions across all the specialties. Although all effort was made to identify and include relevant RCTs, there is a chance of missing relevant studies. This weakness is, however, inherent to all systematic reviews. Furthermore, the reviewed TPE interventions were quite heterogeneous, owing to heterogenous study samples, delivery formats and settings.

Another important limitation is that we only meta-analyzed the primary outcomes presented by RCTs included in this review. This was decided by the review team because the studies focused on biomedical outcomes may not have tailored content to improve other secondary outcomes such as QoL or the psychological health of patients. These latter outcomes were presented by only a few studies. Future studies should consider a meta-analysis of both the primary and secondary outcomes in all the studies.

This review delineates the strategies and content of interventions and their associations with the effectiveness of interventions. However, the findings yielded from the subgroup and meta-regression analyses should be interpreted with caution because of the observational nature of this evidence. We recommend interventionists consider qualitative and process evaluations in the future to identify effective and acceptable approaches. Furthermore, data for patient-level covariates were not extracted in the present review. The duration of intervention (duration and number of sessions and duration of the overall program) is a critical variable; however, these data were very inconsistently reported in the literature and missing in most instances.

## 6. Conclusions

In conclusion, TPE interventions lead to significant improvements across several health indicators among patients with diabetes and obesity. The trials included in this review used heterogeneous delivery techniques and intervention delivery agents. The use of electronic media such as short messaging services (SMS), website-based educational programs and animation media can be used to deliver TPE effectively. Using non-specialist delivery agents and electronic media may be cost-effective and reduce the work burden on physicians.

## Figures and Tables

**Figure 1 nutrients-14-03807-f001:**
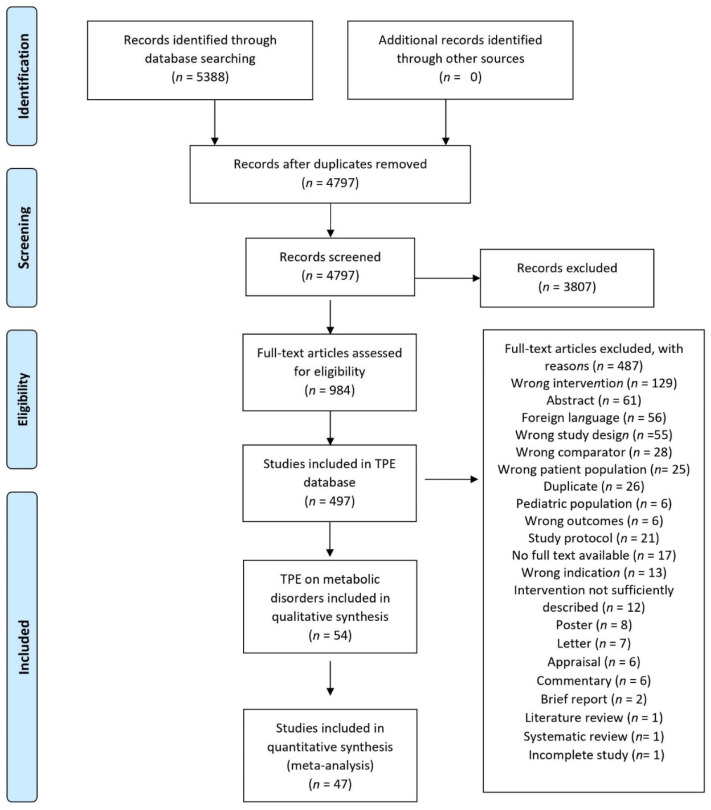
PRISMA flowchart for selection of eligible studies.

**Figure 2 nutrients-14-03807-f002:**
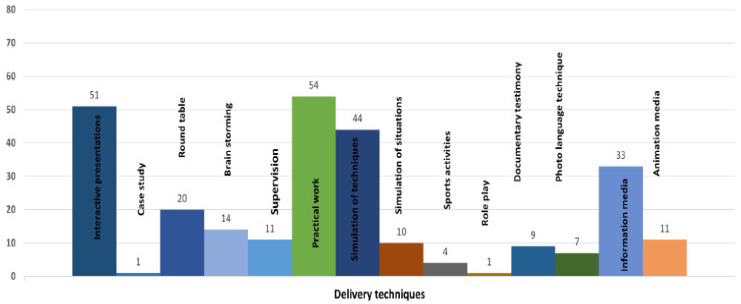
Delivery techniques for TPE interventions.

**Figure 3 nutrients-14-03807-f003:**
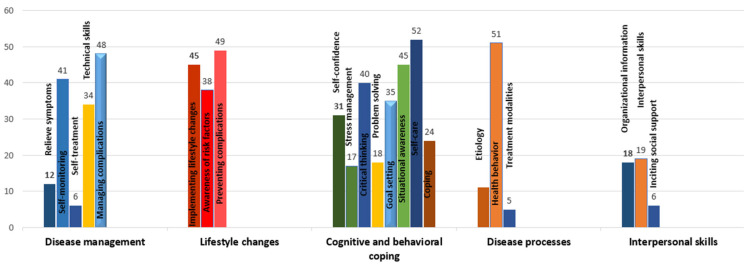
Content and skills for development of TPE interventions.

**Figure 4 nutrients-14-03807-f004:**
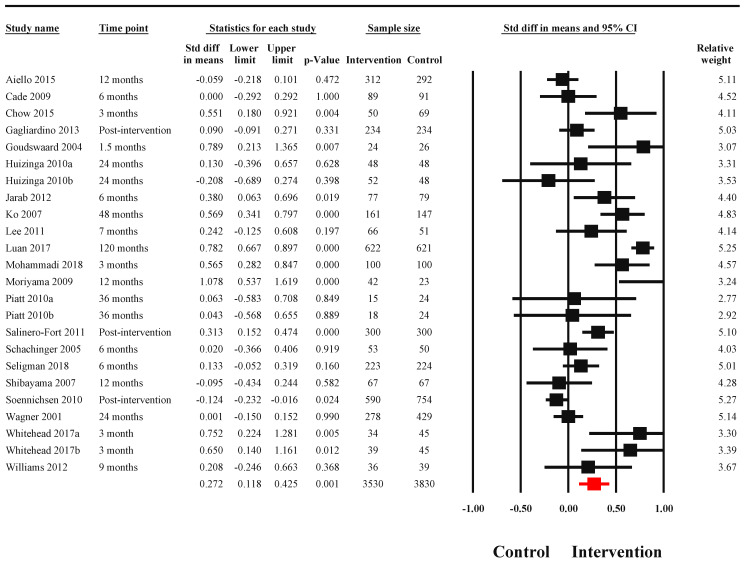
Effectiveness of TPE interventions in improving serum HbA1c levels among patients with diabetes mellitus. The black blocks present point estimates and 95% confidence intervals for individual studies. The red block presents pooled effect size and associated 95% confidence intervals [5,9,28,29,31,35,37,40,41,42,43,46,49,51,52,54,58,59,62,63,70].

**Figure 5 nutrients-14-03807-f005:**
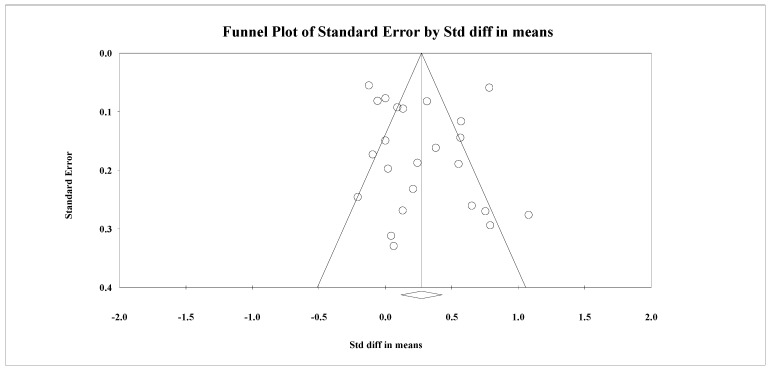
Publication bias in reporting of serum HbA1c levels outcome.

**Figure 6 nutrients-14-03807-f006:**
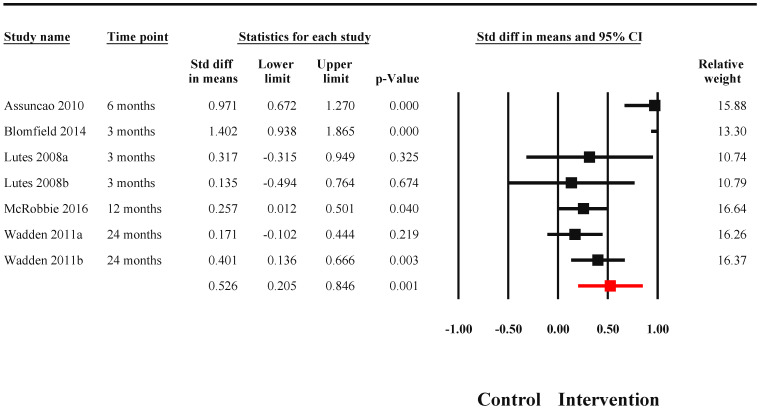
Effectiveness of TPE interventions in improving body weight among patients with obesity. The black blocks present point estimates and 95% confidence intervals for individual studies. The red block presents pooled effect size and associated 95% confidence intervals [25,39,55,64,66].

**Figure 7 nutrients-14-03807-f007:**
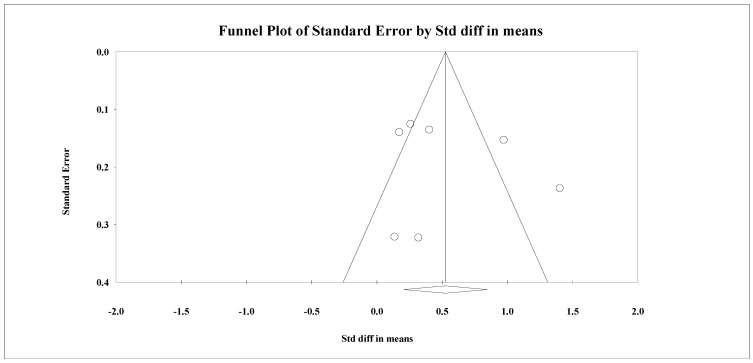
Publication bias in reporting of serum HbA1c levels outcome.

**Figure 8 nutrients-14-03807-f008:**
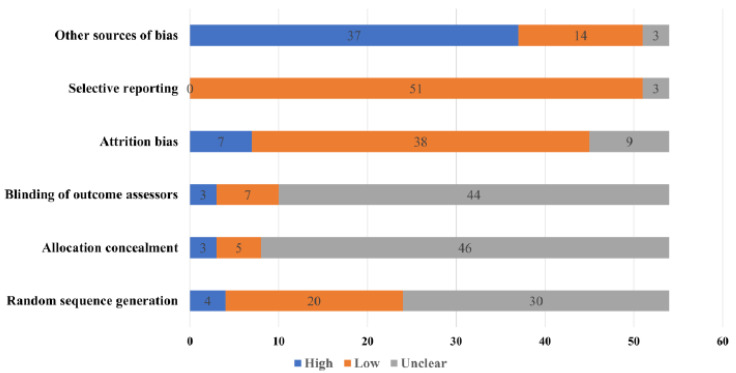
Summary graph for risk of bias among included RCTs.

**Table 1 nutrients-14-03807-t001:** Subgroup analyses for biomedical outcomes (*n* = 38).

Group	n	SMD	95% CI		I-Squared	Q	df	*p*
**Serum HbA1c levels among patients with diabetes**								
**Delivery agents**								
Allied health workers	12	0.26	0.10	0.43	59.98	21.49	3	<0.001
Doctors	1	−0.12	−0.23	−0.02	0.00			
Multidisciplinary	10	0.35	0.09	0.61	92.33			
Peers	1	0.00	−0.29	0.29	0.00			
**Format of delivery**								
Group	9	0.27	0.05	0.50	85.38	2.32	3	0.51
Individual	9	0.33	0.04	0.62	92.08			
Mixed	3	0.28	−0.08	0.64	80.28			
Telephone	3	0.05	−0.23	0.33	0.00			
**Body weight among patients with obesity**								
**Delivery agents**								
Allied health workers	4	0.44	0.11	0.77	83.32	15.02	2	<0.001
Multidisciplinary	2	0.23	−0.22	0.67	0.00			
Research staff	1	1.40	0.94	1.87	0.00			
**Format of delivery**								
Group	1	0.26	0.01	0.50	0.00	19.61	3.00	<0.001
Individual	2	0.23	−0.22	0.67	0.00			
Internet	1	1.40	0.94	1.87	0.00			
Mixed	3	0.51	0.06	0.96	87.21			

**Table 2 nutrients-14-03807-t002:** Association of effect size of TPE interventions (for biomedical outcomes) with content of interventions.

Covariate	B	SE	t	*p*-Value
**Intercept**	−0.402	0.341	−1.18	0.25
**Disease management**	0.061	0.063	0.97	0.34
**Lifestyle**	0.016	0.136	0.11	0.91
**Coping total**	0.009	0.041	0.22	0.83
**Disease processes**	0.19	0.114	1.67	0.11
**Interpersonal skill**	0.15	0.092	1.64	0.12
**R^2^ = 0.37**				

## Data Availability

All reported data are available in the manuscript.

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
