# Peer review of "Effectiveness of Therapeutic Patient Education Interventions in Obesity and Diabetes: A Systematic Review and Meta-Analysis of Randomized Controlled Trials"

_nutrients, 2022, doi:10.3390/nu14183807_

Round 1
Reviewer 1 Report
Thank you for the opportunity to review, Effectiveness of therapeutic patient education interventions in obesity and diabetes: A critical systematic review and meta- analysis of randomized controlled trials.
The authors have undertaken a systematic review and meta-analysis of TPE interventions for diabetes and obesity.
I believe there are a number of issues with this paper that need to be addressed. I recommend major revisions.
1) The authors do not establish that the review they have undertaken is ‘critical’. If the authors wish to maintain this position, they need to clearly justify what makes their manuscript critical.
2) Diabetes mellitus is not a single condition and it is type 2 diabetes that is usually associated with obesity. The authors should be clear as to whether they included articles on both type one and type two diabetes and, if they have included studies of TPE in type one diabetes, this inclusion should be justified within the text. If they have not included studies of type one diabetes, references to diabetes should be replaced with type 2 diabetes and type one diabetes should presumably be noted within exclusion criteria.
3) Prisma guidelines are not always clearly and precisely followed. Some examples of this are noted below. If the authors have not followed Prisma guidelines, they should be clear about this. Equally, if they have followed Prisma guidelines, this should also be clear – and the manuscript should be reworked to more clearly align with the guidelines. I note a number of these below.
4) Relatedly, no mention is made of a study protocol and this should be clarified in either case.
5) The discussion and conclusion appear to go well beyond the bounds of the systematic review and meta-analysis.
6) There are a large number of errors of grammar and word choice. Many are noted below:
Minor issues
p. 2 A couple of minor issues with second sentence on the ‘primary aim’ of TPE-
1) “empower them with resources” – perhaps ‘empower them with knowledge and skills’ might be more appropriate? Resources, particularly in the case of type II diabetes, might include continuous glucose monitoring (or SMBG), as well as pharmaceutical resources.
2) it might be more accurate to rephrase the final section as ‘reduce the likelihood/risks/chances of developing further complications’. Afterall, a patient may follow TPE and find benefits from it, but still end up with complications in later years.
p. 2 Although a PRISMA flow diagram is included later in the manuscript, the authors should be clear at the start of their methods section whether they followed the PRISMA guidelines (I assume they did, given the inclusion of the flow diagram later).
p. 2 Prisma guidelines state that in the eligibility section, details should be provided about how studies were grouped for synthesis. The third and second to last sentences appear to address this, but can you make this more explicit, please?
Also, any exclusions based on language have not been specified.
p. 2 The supplementary table does not appear to have been provided to reviewers. I am unable to review the search strategy fully as a result. The URL for supplementary materials results in a 404 file not found error.
pp2-3 Data extraction does not specify the process by which the two reviewers (or wider study team) checked for agreement between reviewers nor how disagreements were resolved.
p. 3 ‘a’ not needed prior to ‘497’
p. 4 no rationale provided for there being 7 fewer articles in the meta-analysis than in the qualitative synthesis.
p. 5 again, no access to supplementary table S2, so this is unreviewed.
p. 5 3.2 Ingredients of interventions – first sentence requires revision (such as removing ‘while’).
p. 5 3.2 – some of the wording is unclear – with some of content categories only having the highest n reported, while in others all the components within a category have been reported.
p. 5 Fig 3 – bar chart is missing labels specific to each bar.
P. 6 “The participants were in goal-setting...” requires revision
Regular errors in grammar and word choice.
p. 7 it is not clear what ‘pill-box’ refers to.
p. 10 correct ‘findings form’ to ‘findings from’
p. 11 – authors claim that ‘TPE interventions also enable the health professionals to tackle psychosocial aspects of chronic diseases’ – however, the meta-analysis has not shown benefit in terms of psycho-social aspects such as quality of life, nor has it measured whether perceptions of stigma or other psycho-social variables are improved by TPE. So it is not clear that this claim is justified.
P. 11 multiple references are being made to WHO report on PTE from 1998 – can this be fleshed out to include additional and more recent references. The report appears to be grey literature. The authors might consider the growing body of literature on patient-centred care, for example.
p. 11 ‘maybe’ should be ‘may be’
p. 12 claim that ‘physicians should be trained in biopsychsocial dimension of care...’ does not appear to be justified based on this systematic review and meta analysis. Relatedly the claim ‘formation of this therapeutic bond is possible only when physicians are actively involved in therapuetic education programs” is also not justified by this review and meta-analysis. In fact, the authors have earlier specified that there is no notable benefit to biomedical outcomes from physician lead programmes.
Reviewer 2 Report
Correia et al. present a systematic review and meta-analysis on the effectiveness of therapeutic patient education (TPE) among persons with obesity or diabetes. They find significant improvements in ‚biomedical outcomes‘ upon the various interventions, with strong heterogeneity across studies / interventions. The topic is very relevant and interesting, and the paper is well-written. My main concern is that the authors meta-analysed studies that are too heterogeneous in too many ways.
1) Figure 4: The individual outcomes cannot be pooled for different reasons. They are clinically different (body weight vs. blood sugar vs. kidney function), have a different meaning among obese vs. diabetic patients and they are on different scales, i.e. the mean differences between them are not comparable. The outcomes need to analysed separately, overall and among obese vs. diabetic patients, also given that there is massive heterogeneity across the interventions. In this context, it would make a lot of sense to open the study extraction to secondary RCT findings on outcomes such as BMI, glucose, HbA1c or eGFR.
2) While it is OK to ask the question whether TPE interventions have beneficial effects in general, more information is needed on which approaches are effective in which study population regarding which endpoint. If it is not possible to address these questions by meta-analyses a more qualitative type of evaluation of the effective vs. non-effective interventions would be of interest.
3) Further stratification by sex, age, geographical location and patient group (inpatient outpatient) would be of interest if possible
4) The duration of the interventions is critical. Was success more short-term or long-term? Can the studies by stratified by duration?
5) Abstract: It should be specified what the biomedical outcomes are
6) Abstract: The overall effect (‚SMD=0.36, 95% 23 CI: 0.23 to 0.49‘) is not interpretable without context; what’s the outcome, which unit does it have?
7) Line 82: What is meant by tracer disorders? This is a term from nursing that not all readers may be familiar with.
8) Discussion: As the authors state a lack of funding and staff may be the main barrier to TPE interventions. Medical staff is overwhelmed with too many other basic tasks in many countries. Where would staff and funding come from? How can short-term success of TPE interventions be maintained over longer durations outside of clinical settings?
Reviewer 3 Report
The authors have developed an interesting work, with an extensive analysis of the literature and a correct meta-analysis. However, in order to improve the quality of work I recommend the following suggestions:
It would be necessary to explain in more detail tables 1 and 2 and their abbreviations. Table 2 is not clear enough; the methodology used is not explained in the text. Also, other abbreviations such as SMD or QoL are not defined in the text.
In the abstract, the result (SMD= 0.272, 95% 23 CI: 0.118 to 0.525, n=7360) would put it behind “HbA1c levels”
2.1. Information Sources & Search Strategy: a priori is separated
Author Response
We are very grateful to the reviewer for their kind feedback
- Table 1 and Table 2 have now been explored in more detail in methods. It is defined as “Subgroup analyses were run to identify differences in effect sizes of TPE interventions according to type of delivery agents and mode of delivery. Random effects meta-regression analysis was used to analyze associations between effect sizes of TPE interventions with content of interventions (disease management, lifestyle changes, coping skills, disease processes and interpersonal skills.)”
- SMD and QoL have now been defined at the first instance. Quality of life is abbreviated as (QoL) and consistently mentioned across the manscript. SMD has been defined in methods as, “Standardized mean differences (SMD) were calculated for continuous outcomes.” And its abbreviation SMD is now consistently used throughout the manuscript.
- The results for HbA1c have now been correctly positioned.
- The word “a priori has now been correctly written as “apriori”
Reviewer 4 Report
Dear authors,
I found the study interesting and easy to read and to understand. Obesity and diabetes are one of the mains problems of public health. I think that you have done a great effort for reviewing all available literature and to show as much information as possible. I have just a few recommendations:
- Please do not use acronyms in the abstract. For example you use SMD without saying the meaning.
- In point 3.4 Efficacy you say "a total of 31 studies (37 trials)". I do not understand how 31 studies can be 37 trials.
- I do not find the information about table 2 analysis in the methods section.
Kind regards
Author Response
We are very grateful to the reviewer for their kind feedback
- We now avoid the use of acronyms in the abstract. We have defined SMD as standardized mean difference before using its abbreviation.
- We have now corrected the sentence which reads as “A total of 31 studies (37 trial arms, n = 9879 participants). In some studies, there were more than 2 trial arms comparing two or more interventions with control group and hence this note.
- Table 2 has now been detailed in the methods section. It reads as, “Subgroup analyses were run to identify differences in effect sizes of TPE interventions according to type of delivery agents and mode of delivery. Random effects meta-regression analysis was used to analyze associations between effect sizes of TPE interventions with content of interventions (disease management, lifestyle changes, coping skills, disease processes and interpersonal skills.)”
Round 2
Reviewer 1 Report
I commend the authors for their thorough response to comments on their original submission. I have found r.1 to be a much clearer manuscript. The separation of outcomes as per the other reviewers comments has significantly clarified the manuscript. The manuscript needs a careful edit for English language, but is otherwise ready for publication.
Two examples of minor corrections below:
p. 2 ‘prevent further complications’ is in the same sentence that has had ‘knowledge and skills’ replace ‘resources’
p. 3 results – highlighted sentence missing ‘data’ between ‘sufficient’ and ‘for’
Author Response
We are very grateful to the reviewer for their kind feedback. We have now proofread the manuscript for any grammatical errors.
- We have rephrased this sentence as, “The primary aim of TPE is to help people with different disorders understand the nature of their disease and empower them with knowledge and skills. Thus, TPE can help them make informed decisions, self-manage their symptoms and prevent further complications [10].”
- The word “data” has now been added.
Reviewer 2 Report
I thank the author for their revisions. If possible I would ask them to modify the following sentence in the conclusions: "However, this evidence in observational in nature due to the nature of meta-regression analysis." This is a meta-analysis of intervention studies, which is why I do not think the research should be marked as observational.
Author Response
We are very grateful to the reviewer for their kind feedback.
This is indeed correct. However, the observational nature of statistical evidence has been mentioned only for subgroup analyses and meta-regression analyses. This has now been clarified in the manuscript:
“However, the findings yielded from subgroup and meta-regression analyses should be interpreted with caution because of the observational nature of this evidence.”